# Impacts of pre-fire conifer density and wildfire severity on ecosystem structure and function at the forest-tundra ecotone

Xanthe J. Walker[1]*, Brain K. Howard[1], Mélanie Jean[2], Jill F. Johnstone[3], Carl Roland[4], Brendan M. Rogers[5], Edward A. G. Schuur[1], Kylen K. Solvik[5], Michelle C. Mack[1]

**1** Center for Ecosystem Science and Society and Department of Biological Sciences, Northern Arizona University, Flagstaff, Arizona, United States of America, **2** Département de biologie, Université de Moncton, Moncton, New Brunswick, Canada, **3** Institute of Arctic Biology, University of Alaska Fairbanks, Fairbanks, AK, United States of America, **4** National Park Service, Central Alaska Network, Fairbanks, AK, United States of America, **5** Woodwell Climate Research Center, Falmouth, Massachusetts, United States of America

* xanthe.walker@gmail.com

**Data Availability Statement:** All data and R code from this study is archived in the US National Science Foundation-funded Bonanza Creek Long

## Abstract

Wildfire frequency and extent is increasing throughout the boreal forest-tundra ecotone as climate warms. Understanding the impacts of wildfire throughout this ecotone is required to make predictions of the rate and magnitude of changes in boreal-tundra landcover, its future flammability, and associated feedbacks to the global carbon (C) cycle and climate. We studied 48 sites spanning a gradient from tundra to low-density spruce stands that were burned in an extensive 2013 wildfire on the north slope of the Alaska Range in Denali National Park and Preserve, central Alaska. We assessed wildfire severity and C emissions, and determined the impacts of severity on understory vegetation composition, conifer tree recruitment, and active layer thickness (ALT). We also assessed conifer seed rain and used a seeding experiment to determine factors controlling post-fire tree regeneration. We found that an average of $2.18 \pm 1.13$ Kg C m$^{-2}$ was emitted from this fire, almost 95% of which came from burning of the organic soil. On average, burn depth of the organic soil was $10.6 \pm 4.5$ cm and both burn depth and total C combusted increased with pre-fire conifer density. Sites with higher pre-fire conifer density were also located at warmer and drier landscape positions and associated with increased ALT post-fire, greater changes in pre- and post-fire understory vegetation communities, and higher post-fire boreal tree recruitment. Our seed rain observations and seeding experiment indicate that the recruitment potential of conifer trees is limited by seed availability in this forest-tundra ecotone. We conclude that the expected climate-induced forest infilling (i.e. increased density) at the forest-tundra ecotone could increase fire severity, but this infilling is unlikely to occur without increases in the availability of viable seed.

## Introduction

Wildfires have historically been rare at the northern forest-tundra ecotone, but as climate continues to warm, they are expected to increase in frequency and extent [1,2]. In boreal forests,

Term Ecological Research Data Catalog, which is part of EDI Data Portal. R code used for statistical analyses is also archived on github at https://github.com/xanthewalker/Denali_fire Walker, Xanthe; Mack, Michelle C; Johnstone, Jill. 2021. Toklat River Fire in Denali National Park and Preserve: Site level environmental, soil, tree, vegetation, and fire characteristics measured in 2016, Bonanza Creek LTER - University of Alaska Fairbanks. BNZ:786, http://www.lter.uaf.edu/data/data-detail/id/786

**Funding:** This project was supported by funding awarded to MCM, JFJ and EAGS from the NASA Arctic Boreal and Vulnerability Experiment (ABoVE) Legacy Carbon grant NNX15AT71A and the Bonanza Creek Long-Term Ecological Research Program, which is funded by National Science Foundation grant # DEB1636476 and the USDA Forest Service Pacific Northwest Research Station. The funders had no role in study design, data collection and analysis, decision to publish, or preparation of the manuscript.

**Competing interests:** The authors have declared that no competing interests exist.

changing fire regimes can shift ecosystems from a carbon (C) sink to a C source to the atmosphere [3], enhance permafrost degradation [4], and alter understory vegetation composition [5] and successional trajectories [6,7], ultimately changing ecosystem C storage potential [8,9]. The structure and function of tundra ecosystems are likely to be similarly impacted by wildfires, and these impacts may be especially important at the forest-tundra ecotone, where fires can facilitate recruitment of boreal species into areas historically dominated by tundra vegetation [10–12]. Understanding the impacts of wildfire at the forest-tundra ecotone is required to make accurate predictions of the rate and magnitude of changes in boreal-tundra landcover, its future flammability, and associated feedbacks to the global C cycle and climate.

At the global scale, one of the most important functions of boreal and tundra ecosystems is their ability to store C in the soil organic layer (SOL) and permafrost [13,14]. These northern ecosystems store approximately 40% of global terrestrial C [15], but changes to the fire regime threaten to disrupt their long-term C storage [3]. Combustion of the SOL dominates C emissions during fire [16,17] and as such is a key component of net ecosystem C balance [18,19]. Combustion of the insulative SOL also increases soil temperatures and active layer thickness (ALT) which in turn stimulates permafrost degradation and exposes organic C to decomposition [4,20], further accelerating C loss to the atmosphere.

Boreal fire severity, or the proportional loss of above and belowground organic material that occurs as a direct consequence of fire [21], is primarily controlled by fuel availability associated with fine-scale drainage conditions and stand type [22]. In boreal black spruce forests C emissions are highest in intermediately drained landscape positions because these areas accumulate deep SOLs and are also sufficiently dry for combustion [7,17]. Field based estimates of tundra fire severity are rare, but pre-fire C pools and C emissions per unit area reported for a tundra wildfire in Alaska [16] are at the high and low range, respectively of those reported for boreal black spruce forests [23,24]. As such, the severity of wildfires and C emissions might decrease in association with decreasing tree cover and increased SOL across the forest-tundra ecotone.

In addition to the direct effects of fire on C emissions, fires also have the potential to restructure understory plant communities [5,25,26]. In boreal forests, seed availability and vegetative propagules in combination with residual SOL depth can control post-fire understory vegetation composition. Severely burned areas are often colonized by deciduous shrubs, graminoids, and annual forbs, whereas low severity fires promote resprouting of evergreen shrubs and recovery of slow-growing mosses and lichens [27]. The effects of fire can also interact with environmental conditions to determine plant community composition after fire [28–31]. Areas of high drainage that experience severe fires are generally the most susceptible to post-fire changes in boreal plant communities [5,27]. Fire-induced changes in plant community composition may be especially prominent at the forest-tundra ecotone where fire creates suitable seedbeds for boreal plant species to establish in tundra ecosystems [10–12].

The expansion of boreal trees into tundra ecosystems is expected to occur across subarctic and Arctic North America in concert with increasing temperatures. However, there has not been a ubiquitous northward or upslope advance in the range of boreal tree species [32–34], suggesting that additional non-climatic factors, such as disturbance from fire [35], may play an important role in determining the position of the forest-tundra ecotone. For boreal trees to establish in tundra ecosystems, trees must allocate resources to reproductive structures, produce viable seed, and seeds must disperse into a suitable substrate for germination and establishment [36]. Fires can facilitate boreal tree recruitment into tundra by releasing the seeds of semi-serotinous species such as black spruce (*Picea mariana*) and creating safe microsites for establishment via deep burning of the SOL and mineral soil exposure [10–12]. In contrast,

fires can also decrease forest cover through post-fire regeneration failure [34,37,38] or drive state changes in canopy species [39,40].

Here we assessed differences in environmental variables, landscape characteristics, and wildfire severity spanning a forest-tundra gradient within Denali National Park and Preserve in Interior Alaska. We tested the impacts of wildfire and pre-fire tree density on post-fire changes in ecosystem structure and function in a region for which we have detailed pre-fire plant community data [41]. Finally, we combined observations with experiments to assess controls over post-fire seedling establishment. We tested the following hypotheses:

1. Sites located on south facing slopes with high drainage will have higher pre-fire conifer density due to warmer and drier conditions.

2. Fire severity and C emissions will increase in association with increasing tree cover and decreasing SOL thickness across the forest-tundra ecotone because forested regions within the ecotone are in warmer and drier landscape positions and therefore have more flammable vegetation and drier organic soils.

3. Post-fire active layer thickness (ALT) will increase with increasing pre-fire conifer density because of their warmer and drier landscape positions and with lower residual SOL depths due to their low insulative properties.

4. Fire-induced changes in plant composition will occur throughout the ecotone and will be greatest at sites experiencing deep burning of the SOL that exposes suitable substrates for seedling recruitment.

5. Post-fire conifer density will increase with pre-fire conifer density and in sites that experienced deep burning of the SOL resulting in a low residual SOL post-fire. This is because higher pre-fire conifer density supplies seed and deep burning exposes suitable substrates for post-fire establishment.

6. Conifer seed rain will be highest in sites with greater pre-fire conifer density that recently burned because fire releases seeds from the cones of semi-serotinous tree species like black spruce.

7. Experimental addition of seed and the removal of understory vegetation via scarification will increase the establishment of seedlings in both burned and unburned sites by removing the constraints on establishment associated with low natural seed rain and high competition from understory vegetation.

Our observational results contribute to our understanding of topo-edaphic features and fire severity in the forest-tundra ecotone and the impacts of fire and pre-fire conifer density on ALT, understory plant species composition, and natural conifer seedling establishment. Our experimental results highlight the importance of seed availability and safe microsites for seedling establishment following fire.

## Methods

### Study area and site selection

Our study spanned a gradient from tundra to low-density spruce stands in Denali National Park and Preserve (DNP) in Alaska, USA (Fig 1). DNP has a continental climate characterized by long cold winters and short warm summers. Influences of the Alaska Range create a cooler and wetter environment within DNP than much of Interior Alaska [42]. DNP headquarters (6343' N, 14858'W) had a mean annual temperature of -2C and total annual precipitation of

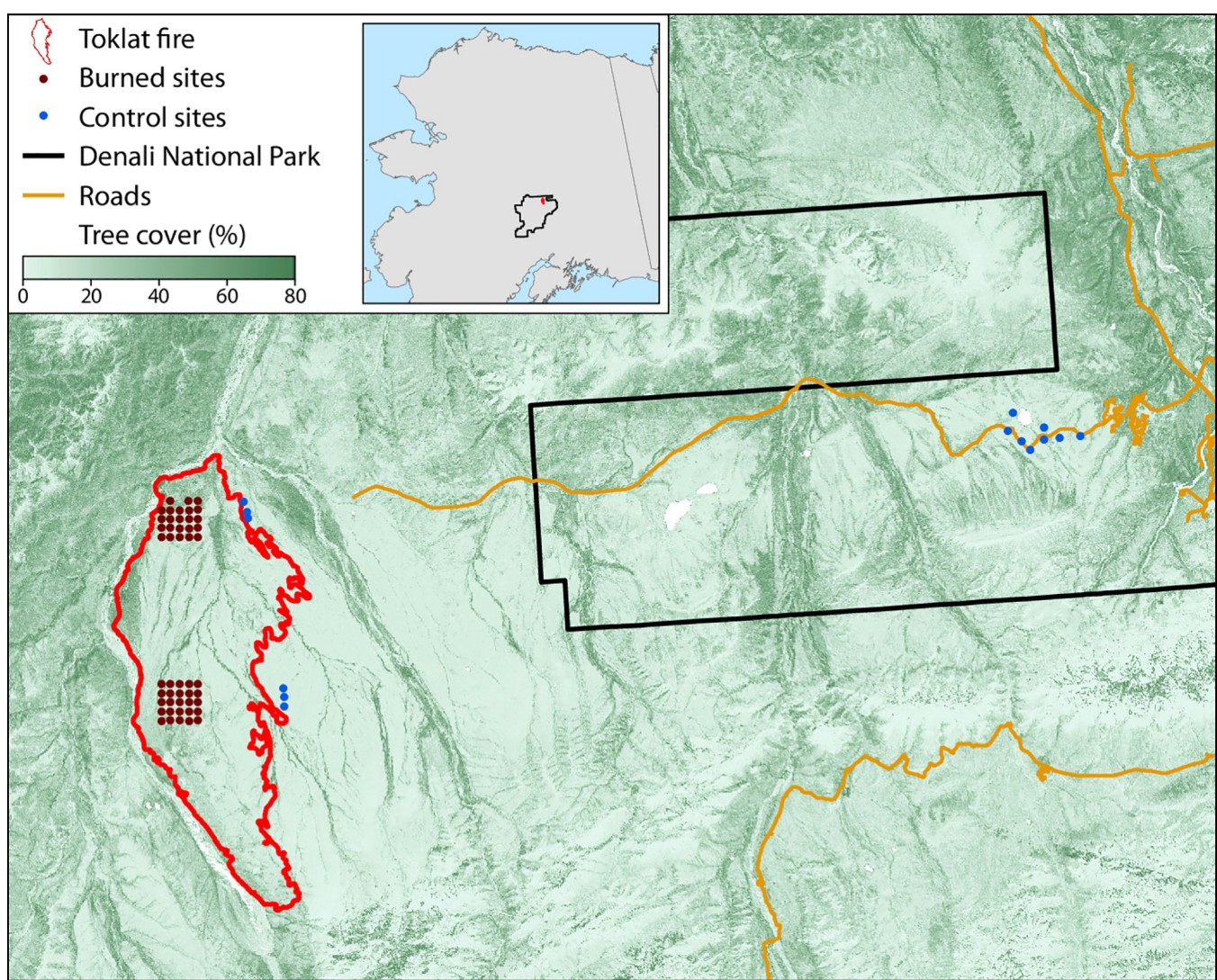

**Fig 1. Map of the study area within the state of Alaska showing locations of burned (red points) and control (blue) sampling sites.** Black line represents the Denali National Park and Preserve boundary (left side is within the Park boundary), orange lines represent major roads, and the red line represents the perimeter of the 2013 Toklat River Fire.

40.4 cm between 1991 and 2020 [43]. Continuous permafrost is common throughout much of DNP, especially in the northern lowlands between watersheds [41]. However, as average air temperatures increase in DNP, the existence of near-surface permafrost is expected to decrease as active layers thicken in the future [4,44].

In the summer of 2013, the East Toklat Fire burned ~34,216 acres in DNP. This fire encompassed two National Park Service Inventory and Monitoring (NPS I&M) grids we call 'East Toklat (ET)' and 'Wigand Creek (WC)'. WC is 10 km north of ET [41]. The two study areas were originally established in 2001 (WC) and 2002 (ET) [41]. Sites are underlain by continuous and discontinuous permafrost supporting a mix of moist acidic tussock tundra interspersed with black spruce and white spruce (*Picea glauca*) forests, with mixed deciduous-conifer forests on warmer south facing slopes. On average, sites in the WC grid had higher pre-fire conifer density ($0.05 \pm 0.01$ stems m$^{-2}$) than sites in the ET grid ($0.01 \pm 0.01$ stems m$^{-2}$), and both grids had sites with no conifer trees present pre-fire. During July 2016, we

resampled both grids following original protocols [45]. Each grid contained 25 plots arranged in a 2 x 2 km area, separated by 500 m. We resampled 48 monitoring plots; two plots were excluded, one lost to a landslide and the other inaccessible due to abundant large dead and downed trees following fire disturbance (WC 23 and 25). Each monitoring plot was 16 m in diameter (~201 m²) and consisted of two perpendicular transects facing each cardinal direction bisecting at 0 m at the plot center.

In addition to the 48 long-term monitoring plots resampled within the burn, 15 control plots representing unburned or pre-fire vegetation were established and sampled following NPS I&M protocols. A total of six control plots were established east of the three southeastern plots within each mini-grid, and were oriented south to north, and separated by 500 m. An additional nine control plots were established ~40 km east, along Stampede Road near Healey, AK, to calibrate our methods for assessing burn depth. Control plots were selected through a GIS approach using the DNP Land Cover Map. Cover types within the burned study sites were weighted by their distribution and then within these selected cover types, tree cover and aspect were selected from their weighted distributions. Control plots were then chosen to have the same landcover within ± 3% tree cover, with matching slopes and aspects to conditions characterized within the burned monitoring plots.

In each plot, we measured slope, aspect, elevation, latitude, and longitude. Slope, aspect, and latitude were used to calculate equivalent latitude as a metric of solar insolation and landscape position corresponding to the position on a sphere that parallels the original slope conditions [46]. Soil monoliths and topographic factors were used to assign a potential site moisture class, as described in [47]. Plots ranged from subxeric-mesic to subhygric among four categories. Only two burned plots were in the subxeric-mesic category, we therefore reclassified them as mesic for subsequent analyses, resulting in three moisture categories.

## Soil sampling

Soil sampling occurred in July 2016. Four organic soil monoliths were collected at every plot. Soil sampling points were located 1 m outside plot perimeters. Monoliths were 10 x 5 x variable depth (cm) and included the entire soil organic layer (SOL) if the active layer thawed to mineral soil or included the depth to frozen ground. Residual SOL depth and active layer thickness (ALT) were measured for each soil pit. Note that these ALT measurements do not represent maximum seasonal thaw as they were taken in July. Monoliths were wrapped in aluminum foil and kept chilled in iced coolers until transported to University Alaska Fairbanks where they were frozen until being shipped to Northern Arizona University (NAU) for analysis.

At an additional eight sampling points, located 1 m outside the perimeter of each plot, we measured SOL depth by making a small cut with a knife and identifying the interface between the organic and mineral soil horizons or frozen ground by visual inspection. At each of these points, we also measured ALT from the SOL surface to frozen ground. In combination with our soil sample measurements (4 per plot), we measured SOL depth and ALT at a total of 576 locations in 48 burned plots (12 points per site). In 23 plots, the residual SOL measurement hit mineral soil in all 12 samplings locations. At four burned plots (ET4, ET12, ET24, WC3) the residual SOL depth measurements were always to frozen ground and not mineral soil. In the remaining 19 plots, there were 58 sampling points where the residual SOL measurement hit frozen ground. This resulted in 470 measurements in 44 sampling plots where we could assess the impact of residual SOL depth on ALT (see statistical analysis).

To estimate burn depth we used the "tussock crown" (as per Mack *et al*. 2011) or "adventitious root" method (as per Boby *et al*. 2010 and Walker *et al*. 2018). In plots where *Eriophorum*

*vaginatum* tussocks were present (n = 40), we located the five nearest tussocks to the soil sample location and measured tussock crown height (TCH) above the residual SOL surface. In plots where *Picea mariana* trees with adventitious roots were present (n = 31), we located the five closest trees to the soil sample location and measured the adventitious root height (ARH) above the residual SOL surface. The same measurements were completed in unburned plots, except either the depth of the adventitious roots below the SOL surface or the TCH above the SOL surface was measured. Burn depth is equivalent to either the ARH or TCH above the residual SOL plus an ARH or TCH offset associated with adventitious root depth below the surface SOL or TCH above the surface SOL measured in unburned plots. In unburned plots, the mean adventitious root depth below the green moss was 2.19 ± 0.06 cm and the mean tussock height was 11.51 ± 0.06 cm. We added 2.19 to all ARH measurements and subtracted 11.51 from all burned TCH measurements to obtain an estimate of burn depth. In a subset of plots (n = 25), both TCH and ARH measurements were made and burn depth was calculated as the average of the two burn depth estimates. We calculated burn depth for each soil profile if tussocks or trees were within ~1 m of the soil sampling location. In 37 plots we were able to estimate burn depth for all four sampling locations, in five plots we calculated three burn depth measurements, in three plots we made two measurements and in one plot we estimated only one burn depth. In two plots (WC20, ET12) no tussocks or trees were present and no burn depth measurements were estimated. In total, burn depth was estimated in 170 sampling locations nested within 46 plots.

## Tree sampling

In 2016, we measured the diameter at breast height (DBH) at the standard height of 1.4 m from the base for all trees ≥ 1.4 m in height and the basal diameter of all trees < 1.4 m that were originally rooted in the plot. Fallen trees that were killed by fire were included in this census. We also assessed tree combustion, where each tree was ranked from 0 to 3; 0 = none, alive and no biomass combusted; 1 = low, only needles/leaves consumed; 2 = moderate, all foliage and majority of fine branches combusted; 3 = high, most of the aboveground canopy including foliage, branches, and bark combusted.

## Understory vegetation communities

Vegetation community data was originally collected at the time of plot establishment in 2001 and 2002 [41,42] and post-fire data was collected in July 2016. Nomenclature followed Hulten (1963) [48] and others, and species present are listed in S2 Table. Along each transect in each plot, vegetation and substrate cover were measured using the line-point intercept (LPI) method. A densitometer attached to the line point apparatus was used to record vegetative cover > 4 m. Below 4 m, a pin was dropped, and plant species were recorded as the pin intercepted plants within strata from 3 to 4 m, 2–3 m, 1–2 m, 0.5–1 m, 30–50 cm, 20–30 cm, 10–20 cm, and 0–10 cm at every point. Starting at the west transect in each plot, a pin was dropped every 50 cm along the entire west-east transect (n = 32 pin drops). The north-south transect was then sampled every 50 cm, skipping points between 6 and 10 m to avoid oversampling at the plot center [45] (n = 24 pin drops). Species % cover was calculated based on the transect data. Cover data obtained through this method tend to be skewed towards abundant species with a high cover, and not be fully representative of the total species richness inside the plot. Therefore, species frequency was assessed both within the four subplots of the 4 $m^2$ quadrat (0, 25, 50, 75 or 100%) and within the entire 200 $m^2$ plot (presence), which yielded a full species occurrence list for species not hit during LPI. These additional species were added to the species cover data by attributing them a small cover value (0.5%).

## Natural regeneration

Post-fire seedling regeneration was assessed within four, 4 $m^2$ quadrats in each plot. One quadrat was placed along the transect in each of the four quadrants. This resulted in an area of 16 $m^2$ where all seedlings were identified to species and counted. Because differentiating white and black spruce seedlings is extremely difficult at this stage of development, we assumed that all seedlings were black spruce unless there was obviously white spruce present pre-fire. Due to the difficulties with identifying spruce seedlings, we pooled seedling counts of black and white spruce for subsequent analyses.

## Seed rain

Spruce seed rain was assessed adjacent to 12 plots (6 burned and 6 control plots). We used the three most southeastern burned plots of both minigrids (plots 1, 6, and 11) and the newly established control plots located ~5 km east of each of these plots. We established a 20 m north-south transect 16 m directly east of these plots. Natural seed rain was determined by randomly placing 10 seed traps along the transect. Seed traps were greenhouse flats (54 x 28 x 6 cm) lined with astroturf (to provide a rough surface with good drainage) and affixed to the ground with four 10 cm nails. Seed traps were placed during July 2016 and collected in August 2016, June 2017, August 2017, June 2018, and August 2018. Total spruce seeds were counted for each plot in each collection period.

## Seeding experiment

Using the same 12 plots where we assessed seed rain, we also established a black spruce seeding experiment. Along the 20 m transects, we randomly established five blocks, each one consisting of four 0.5 x 0.5 m quadrats. Blocks and quadrats were marked by stringing yarn around wooden skewers outlining each quadrat. Half of our treatments received an artificial soil scarring by removing living vascular and non-vascular species in order to mimic the immediate post-fire environment. Within each block, we randomly assigned quadrats to one of four treatments: i) not seeded and not scarified (control), ii) not seeded and scarified, ii) seeded and not scarified, or iv) seeded and scarified. Each of the four treatments was replicated 60 times: 30 times in burned plots and 30 times in control plots (n = 240 quadrats). Prior to application of seeding or scarification treatments, natural spruce seedling regeneration was assessed and removed. In August 2016, we seeded an estimated 250 black spruce seeds (0.187 g) in each of the 120 quadrats that were seeded (~30,000 seeds). Seeds for the seeding experiment were collected during July 2016 within a 5 km radius of each grid, and on average ~16% of collected seeds were viable (S1 File–Seed collection and germination trials). Counts of germinated seeds were completed in August 2017 and August 2018.

## Laboratory soil sampling

To assess the C content of the collected SOL monoliths we followed standard protocols [16,17,49]. In the laboratory, we thawed soil monoliths at room temperature (~25˚C) for approximately 24 hours. We bisected the monoliths depth-wise using an electric carving knife and one half of the monolith was re-frozen for archival purposes. With the remaining half, all live moss or vascular plants were sliced off and discarded from monoliths collected in burned plots. For unburned monoliths, the dimensions of living material (green moss) were measured and retained for further analysis. The remainder of all monoliths were divided into 5 cm depth increments, with the last sample of variable depth depending on the location of the organic to mineral soil or frozen ground interface. Two measurements each of depth, height, and width

were obtained from each increment and the wet weight was recorded. The live material from unburned stands and the 5 cm increment samples from all monoliths were homogenized by hand and any rocks, sticks, or coarse roots >2 mm in diameter were removed. Fine (< 2 mm) and coarse (>2 mm) organic fractions were weighed wet and dried at 60°C for 48 hours to determine dry matter content. Rock volume was estimated by water displacement in a graduated cylinder. Fine organic fractions were ground and the percent C was determined using a Costech Elemental Analyzer calibrated with the NIST peach leaves standard. We determined the bulk density of fine organic fractions for each sample by dividing the dry weight by the sample volume excluding rock volume. We calculated C content by multiplying the sample depth by the bulk density and percent C of the sample.

## Data analysis

All analyses were conducted in the statistical software program R version 4.0 [50]. Data arrangements and basic calculations were done using R packages 'dplyr' [51] and 'tidyr' [52]. For all mixed models that follow, the significance of fixed effects, both interactions and main effects, were assessed using likelihood ratio tests of the full model against reduced models and verified using AIC [53]. Model reduction only occurred on interaction terms. We present results from the fullest additive model, not the minimum adequate model, so that we could test the impact of our hypothesized predictors outlined in the introduction. We verified that the statistical assumptions of homogeneity of variance and independence were not violated by visually inspecting residual versus fitted values, all explanatory variables, and each grouping level of the random intercepts [53]. Model results were plotted using the 'effects' [54] and 'ggplot2' [55] R packages. For ease of visualizing the results, we grouped sites into the following tree density categories with equal number of sites in each group: low density = 0 stems m$^{-2}$, medium density = 0.0001 to 0.0249 stems m$^{-2}$, and high density = >0.0249 stems m$^{-2}$.

**Topo-edaphic gradients.** Stem counts and diameter measurements were used to calculate tree density (number stems m$^{-2}$). We fit linear mixed effects models (LMMs) using the R package 'nlme' [56] to test how equivalent latitude and moisture category impacted pre-fire conifer density (stems m$^{-2}$). We included a random effect of grid to account for potential spatial non-independence of sites located within the same grids and a variance structure (VarConstPower) to account for changing variance with latitude that was apparent in the model residuals [53].

**Aboveground and belowground carbon combustion.** Stem counts, diameter measurements, and published allometric equations [57] were used to calculate aboveground biomass (kg dry matter m$^{-2}$) of the total tree, bark, main branches, fine branches, and needles/leaves, for each tree species in each plot. Total tree biomass combusted was calculated for each tree from the assigned combustion class and affected biomass components (foliage, branches, and bark). We summed individual tree estimates and divided by the sample area to estimate pre-fire biomass and biomass combustion (kg dry matter m$^{-2}$) on an area basis in each plot. We assumed a biomass C content of 50%.

We modelled soil C content as a function of depth using soil monoliths from unburned plots. For each soil monolith (n = 50), we calculated the cumulative sums of C content by 5 cm depth increments starting from the surface (250 depth increments). We fit LMMs with fixed effects for depth, depth squared, moisture category, and the interaction between moisture and depth. Models had a hierarchical random effects structure of plot (15 plots) and soil monolith (50 monoliths) nested within plot to account for the non-independence of soil depth increments sampled within a soil monolith and the spatial non-independence of soil monoliths located within a plot. We also included a variance structure (VarConstPower) to account for increasing variance with depth [53].

We found that soil C content was significantly impacted by depth squared and the interaction between depth and moisture (S1 Table). We used these models to predict carbon combustion (kg C m$^{-2}$) based on burn depth for each burn depth measurement (n = 170). We calculated pre-fire SOL C pools by summing the measured residual C pools and the modelled combusted C pools, which were then averaged per plot. Using the plot level estimates of aboveground and belowground pre-fire C pools and C combusted we calculated total C combusted (g C m$^{-2}$) (aboveground + belowground C combusted), the proportion of total C combusted (total C combusted/total pre-fire C pool), and the proportion of total C combusted attributed to the belowground component (total belowground C combusted/total C combusted).

**Burn depth, carbon combustion, and active layer thickness.** Using burn depth estimates (170 measurements: 46 plots x 2 to 4 measurements per plot), we calculated pre-fire SOL depth (burn depth + residual SOL) and the proportion of SOL that combusted (burn depth/ pre-fire SOL depth). We modelled burn depth, residual SOL depth, and proportion of SOL combusted as dependent variables related to pre-fire conifer density, pre-fire SOL depth, and the interaction between pre-fire conifer density and SOL depth using LMMs. We included random effects of grid (2 levels) and plot (46 levels) nested within grid to account for the spatial non-independence of measurements.

We also used LMMs to model average plot (n = 46) total C combusted (g C m$^{-2}$) as a function of total C pre-fire, pre-fire conifer density, and the interaction between total C pre-fire and pre-fire conifer density. We used generalized linear mixed models (GLMM) in the package 'lme4' [58] with a Gamma error distribution and log link function to model the proportion of total C combusted as function of total C pre-fire and pre-fire conifer density. For these models, we used a random effect of grid (2 levels).

We used GLMM with same error distribution as above to model ALT (470 observations: 44 plots x 8 to 12 measurements per plot) as a function of residual SOL depth, pre-fire conifer density, and the interaction between residual SOL depth and pre-fire conifer density. We included random effects of grid (2 levels) and plot (44 levels) nested within grid to account for the spatial non-independence of measurements.

**Understory vegetation communities.** The Bray-Curtis distance was applied to pre- and post-fire species composition data [59]. Community composition was compared pre- and post-fire and according to the three spruce density levels with a multivariate analysis of variance (PerMANOVA; 5000 permutations) using the 'adonis' function from the package 'vegan' with grid point as a random effect (strata) in R [60,61], followed by pairwise post-hoc tests where appropriate [62]. Spruce density was categorized as low = 0 stems m$^{-2}$, medium = 0.0001 to 0.0249 stems m$^{-2}$, and high = >0.0249 stems m$^{-2}$. Multivariate dispersion was verified with the 'betadisper' function from the 'vegan' package. It differed significantly between the pre- and post-fire groups (p<0.05), but not among spruce densities (p>0.05). Differences in multivariate dispersion can lead to confusion between within-group variation (dispersion) and group mean values. PerMANOVA is less sensitive to this issue than some of its alternatives [63], so we proceeded with the analyses.

Results were graphically represented with a nonmetric multidimensional scaling (NMDS) ordination [64]. The best NMDS solution was selected based on 20 independent runs with 100 iterations and selected the number of axes based on stress values according to dimensionality [64]. We calculated a pre- and post-fire species turnover index (ranging from 0–1, where 0 is no change in species composition and 1 is a complete change in species composition) at each of the grid points using the 'turnover' function in the 'codyn' package [65,66]. This index converts the cover data into presence-absence to calculate species turnover. We then used a LMM with a random effect of grid to assess how species turnover was impacted by residual SOL depth, burn depth, and pre-fire conifer density. We did not include interactions between these

fixed effects because we were primarily interested in the main effects and had insufficient sample size to reliably test interactions. Model diagnostics and the significance of fixed effects were assessed as described above.

**Natural spruce regeneration.** We modelled the total abundance of white and black spruce seedlings at 46 plots using GLMMs with fixed effects of pre-fire tree density, residual SOL depth, and burn depth, and a random effect of grid in the package 'glmmTMB' [67]. We did not include residual SOL surface bulk density despite its potential influence on spruce regeneration because it covaried with residual SOL depth (*correlation* = -0.65, p-value<0.001). We did not include interactions between fixed effects; we were primarily interested in the main effects and we had insufficient sample size to test interactions. As expected, numerous plots had a spruce seedling density of zero. To assess if zero-inflation was necessary and determine the correct error distribution we built GLMMs with: 1) a Poisson distribution where the variance is equal to the mean, 2) a type I negative binomial distribution where the variance increases linearly with the mean, and 3) a type II negative binomial distribution where the variance increases quadratically with the mean [68]. For every distribution, we also modeled the presence or absence of zero inflation. We included grid (2 levels) as a random intercept. We used AIC to choose the best model. Spruce seedling abundance was modeled with a type II negative binomial distribution and model diagnostics and the significance of fixed effects were assessed as described above.

**Seed rain.** To assess how seed rain at 12 plots (60 observations: 12 plots x 5 collections) was impacted by tree density and recent fire we used GLMM, with fixed effects of tree density, stand burn status (burned or control), and their interaction. We included random effects of collection period (5 levels) crossed with grid (2 levels) to account for spatial and temporal non-independence of seed rain counts. Although burned and unburned plots were not spatially independent from one another within each grid, the variance associated with this grouping factor was essentially zero, preventing model convergence, and was therefore not included. We chose an error distribution and determined if zero-inflation was needed as described above. Seed rain was modelled with a type II negative binomial distribution. Model diagnostics and the significance of fixed effects were assessed as described above.

**Seeding experiment.** We used a GLMM to test if seedling counts (480 observations: 240 quadrats x 2 years) varied with seeding (seeded or not), burn (unburned or burned) and scarification (scarred or no scarification), including interactions between these variables. We included random effects of sampling year (2 levels) crossed with grid (2 levels), plot (12 levels) nested within grid, and block (5 levels) nested within plot, burn status, and grid to account for temporal and spatial non-independence of seedling counts. The spatial non-independence of burned and unburned plots nested within each grid were not included for the same reason described above. We chose an error distribution and determined if zero-inflation was needed as described above. Seedling count was modelled with a type I negative binomial distribution. Model diagnostics and the significance of fixed effects were assessed as described above.

## Results

### Topo-edaphic gradients

Pre-fire conifer density (stems m$^{-2}$) was strongly related to both equivalent latitude and moisture category (Fig 2 and Table 2). As equivalent latitude (a metric that accounts for slope, aspect, and latitude) increased, density decreased. In all sites classified as subhygric there were no conifer trees pre-fire. Conifer trees were present in mesic and mesic-subhygric sites, but most abundant in mesic sites.

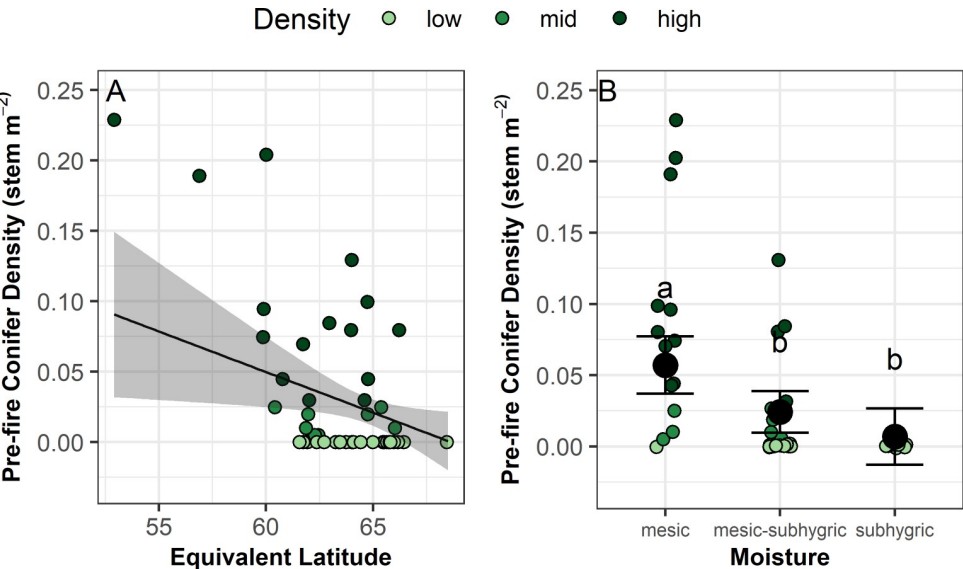

**Fig 2. Pre-fire conifer density (stems·m$^{-2}$) as a function of equivalent latitude and moisture category.** In a) the black line represents modelled relationships from a linear mixed effects model with random effects of grid (2 levels). Shading represents 95% confidence regions. In b) the black circles and error bars represent model fit and 95% confidence intervals. See Table 2 for model results.

## Burn depth, carbon combustion, and active layer thickness

Mean burn depth of the SOL was 10.6 ± 4.5 cm with a range of 0 to 21.5 cm. We found that burn depth slightly increased with both pre-fire SOL depth and conifer density (Fig 3 and Table 2). Residual SOL depth was strongly predicted by pre-fire SOL depth, highlighting a relatively consistent burn depth across all plots, and slightly decreased with increasing pre-fire conifer density (Fig 3 and Table 2). The proportion of SOL combusted strongly decreased with pre-fire SOL depth and slightly increased with conifer density (Fig 3 and Table 2).

We found an average total C combustion of 2.18 kg C m$^{-2}$ (Table 1). The vast majority (~94%) of this was from belowground SOL combustion. On average, belowground C combustion was 1.96 kg C m$^{-2}$ and aboveground was 0.32 kg C m$^{-2}$. Both total C combustion and the proportion of total C combusted increased with pre-fire conifer density (Fig 4 and Table 2). Higher pre-fire C pools resulted in higher total C combustion and a decrease in the proportion of total C combusted (Fig 3 and Table 2).

Both residual SOL depth and conifer density significantly impacted post-fire ALT (Fig 5 and Table 2). ALT was greatest when residual SOL depth was lower and when conifer density was higher.

## Understory vegetation communities

Understory plant community composition varied in response to fire ($F_{1,92} = 33.43$, $p<0.0001$, $R^2 = 0.25$) and pre-fire conifer density ($F_{2,92} = 4.69$, $p<0.0001$, $R^2 = 0.07$). These patterns were illustrated on the NMDS ordination, where sites approximately sorted according to pre- vs. post-fire communities along the first axis (NMDS1), and to a lesser extent according to conifer density along the second axis (NMSD2) with tundra dominated by tussock sedges in the lower section and higher conifer densities in the upper section (Fig 6A; S2 Table). Differences in species composition were found between the low spruce density and both the mid and high spruce density groups which were similar to each other (pairwise tests p<0.05; Fig 6A). The low

**Table 1. Summary of measured and estimated burned plot characteristics.**

| Variable | Units | Mean ± SD | Range | # |
|---|---|---|---|---|
| Elevation | m.a.s.l. | 527.71 ± 47.63 | 453–606 | 48 |
| Slope | radians | 4.08 ± 4.08 | 0–23 | 48 |
| Aspect | radians | 203.23 ± 103.47 | 1–135 | 48 |
| Thaw depth | cm | 51.15 ± 22.34 | 12–130 | 576 |
| Pre-fire black spruce density | stems·m$^{-2}$ | 0.02 ± 0.05 | 0–0.23 | 48 |
| Pre-fire white spruce density | stems·m$^{-2}$ | 0.01 ± 0.02 | 0–0.1 | 48 |
| Conifer seed rain | seeds m$^{-2}$ year$^{-1}$ | 2.32 ± 3.06 | 0.22–8.01 | 30 |
| White spruce seedling density post-fire | stems·m$^{-2}$ | 0.02 ± 0.09 | 0–0.56 | 48 |
| Black spruce seedling density post-fire | stems·m$^{-2}$ | 0.14 ± 0.28 | 0–1.25 | 48 |
| Pre-fire aboveground carbon pool | kg·C·m$^{-2}$ | 0.45 ± 0.82 | 0–3.16 | 48 |
| Aboveground carbon combusted | kg·C·m$^{-2}$ | 0.32 ± 0.56 | 0–2.32 | 48 |
| Pre-fire soil organic layer (SOL) depth | cm | 27.95 ± 9.71 | 4.82–54.62 | 170 |
| Burn depth | cm | 10.57 ± 4.48 | 0–21.48 | 170 |
| Residual SOL depth | cm | 16.06 ± 9.83 | 0–51 | 470 |
| Residual SOL surface bulk density | g·cm$^{-3}$ | 0.14 ± 0.07 | 0.04–0.43 | 170 |
| Proportion of soil organic layer combusted | - | 0.43 ± 0.23 | 0–1 | 170 |
| Pre-fire belowground carbon (C) pool | kg·C·m$^{-2}$ | 8.18 ± 4.63 | 1.53–24.92 | 170 |
| Belowground C combusted | kg·C·m$^{-2}$ | 1.96 ± 1.09 | 0–4.94 | 170 |
| Total pre-fire C pool | kg·C·m$^{-2}$ | 8.42 ± 3.03 | 3.07–15.52 | 46 |
| Total C combusted | kg·C·m$^{-2}$ | 2.18 ± 1.13 | 0.32–5.94 | 46 |
| Proportion of total C combusted | - | 0.28 ± 0.14 | 0.05–0.66 | 46 |
| Proportion of total C combusted from SOL | - | 0.94 ± 0.11 | 0.56–1 | 46 |

spruce density group was characterized by an abundance of the tussock-forming sedge *Erio-phorum vaginatum* (Fig 6B). Higher spruce density sites were characterized by species with boreal affinities, such as *Picea mariana*, *Rhododendron groenlandicum*, and *Myrica gale* (Fig 5B). Most sites moved horizontally across the NMDS space after fire, showing a change in species composition from pre-fire communities dominated by small ericaceous (*Vaccinium uliginosum*, *Vaccinium vitis-idaea*, and *Rhododendron tomentosum* subsp. *decumbens*) and deciduous (*Betula nana*) shrubs and *Carex bigelowii* towards communities strongly associated with the presence of fireweed (*Chamerion angustifolium*) and grasses like *Calamagrostis lapponica*. (Fig 6; S2 Table). This change in community composition was slightly larger in the high than the low spruce density group. Lesser change was observed along the forest-tundra gradient (Fig 6).

Species with the highest negative change were associated more strongly with pre-fire communities at low spruce density sites, while species that increased in frequency were fire-adapted species characteristic of our post-fire communities (Fig 7). However, while we observed large post-fire changes in community composition based on cover data, about half of the species (48/102) had a mean difference in frequency that was 0 ± 0.01%. Indeed, we found that the average species turnover was 0.34 ± 0.02, indicating relatively low changes in terms of species occurrence at the sites. Species turnover was highest in areas with a lower residual SOL depth but was not impacted by burn depth or pre-fire conifer density (Fig 8 and Table 2).

### Natural spruce regeneration

Across all burned plots, black spruce mean seedling density was 0.14 stems m$^{-2}$ (range 0–1.25) and white spruce was 0.02 stems m$^{-2}$ (range 0–0.56). We found no evidence of recruitment in any of the unburned control plots. The majority (54%) of burned sites also had no spruce

**Table 2. The effects of environmental variables, landscape characteristics, wildfire severity, and pre-fire tree density on post-fire changes in ecosystem structure and function.**

| Model | Variable | Estimate | Std. Error | Z or t-value | $R^2_m$ ($R^2_c$) |
|---|---|---|---|---|---|
| Pre-fire conifer density | Intercept (moisture: mesic) | 0.42 | 0.15 | 2.69 | 0.52 (0.52) |
| | **Equivalent latitude** | **-0.03** | **0.01** | **-2.81** | |
| | **Moisture: mesic-subhygric** | **-0.05** | **0.01** | **-3.68** | |
| | **Moisture: subhygric** | **-0.005** | **0.002** | **-2.36** | |
| Burn depth | Intercept | 6.74 | 1.05 | 6.45 | 0.12 (0.75) |
| | **Pre-fire SOL depth** | **0.11** | **0.03** | **4.25** | |
| | **Pre-fire conifer density** | **23.38** | **10.23** | **2.28** | |
| Residual SOL depth | Intercept | -6.74 | 1.05 | -6.44 | 0.80 (0.94) |
| | **Pre-fire SOL depth** | **0.89** | **0.03** | **33.14** | |
| | **Pre-fire conifer density** | **-23.39** | **10.27** | **-2.28** | |
| Proportion SOL combusted | Intercept | 0.77 | 0.05 | 17.23 | 0.42 (0.74) |
| | **Pre-fire SOL depth** | **-0.01** | **0.001** | **-11.21** | |
| | **Pre-fire conifer density** | **0.77** | **0.35** | **2.24** | |
| Total C combusted | Intercept | 0.67 | 0.47 | 1.44 | 0.17 (0.17) |
| | **Pre-fire C** | **0.14** | **0.05** | **2.87** | |
| | **Pre-fire conifer density** | **8.96** | **2.63** | **3.41** | |
| Proportion of total pre-fire C combusted | Intercept | -0.98 | 0.20 | -4.87 | 0.21 (0.21) |
| | **Pre-fire C** | **-0.05** | **0.02** | **-2.56** | |
| | **Pre-fire conifer density** | **3.20** | **1.13** | **2.84** | |
| Active layer thickness | Intercept | 3.98 | 0.06 | 64.90 | 0.11 (0.40) |
| | **Residual SOL depth** | **-.006** | **0.001** | **-4.49** | |
| | **Pre-fire conifer density** | **1.08** | **0.44** | **2.42** | |
| Species turnover | Intercept | 0.42 | 0.06 | 6.72 | 0.17 (017) |
| | **Residual SOL depth** | **-0.004** | **0.002** | **-2.37** | |
| | Burn depth | 0.002 | 0.004 | -0.57 | |
| | Pre-fire conifer density | 0.21 | 0.23 | 0.93 | |
| Natural spruce regeneration | Intercept | 0.34 | 1.57 | 0.22 | 0.15 (0.76) |
| | Residual SOL depth | -0.01 | 0.03 | -0.40 | |
| | Burn depth | -0.02 | 0.06 | -0.24 | |
| | **Pre-fire conifer density** | **11.16** | **5.02** | **2.22** | |
| Seed rain | Intercept (control) | -0.64 | 0.75 | -0.85 | 0.09 (0.51) |
| | Burned | -0.08 | 0.37 | -0.22 | |
| | **Pre-fire conifer density** | **22.78** | **7.33** | **3.10** | |
| Experimental spruce regeneration | Intercept (no seed, control) | -4.40 | 0.79 | -5.58 | 0.88 (0.99) |
| | **Seeded** | **5.52** | **0.71** | **7.75** | |
| | **Burned** | **1.99** | **0.88** | **2.25** | |
| | **Scarification** | **0.62** | **0.08** | **7.41** | |
| | **Seeded * Burned** | **-1.83** | **0.78** | **-2.35** | |

Each of the response variables (Model) and fixed effects (Variables) are associated with the hypotheses outlined in the introduction. Estimates (± std.error) represent effect size relative to the intercept based on mixed effect models with model-dependent levels of random intercepts (see methods). Marginal $R^2$ ($R^2_m$) provides the variance explained only by fixed effects and conditional $R^2$ ($R^2_c$) provides the variance explained by the entire model. See Table 1 for units and number of measurements associated with each response variable. Bolded variables indicate significant fixed effects. See Figs 1–4 and 7 for graphical depiction of results.

regeneration post-fire. We found that total conifer (white and black spruce) regeneration was not impacted by residual SOL depth or burn depth and slightly increased with pre-fire conifer density (Fig 9 and Table 2).

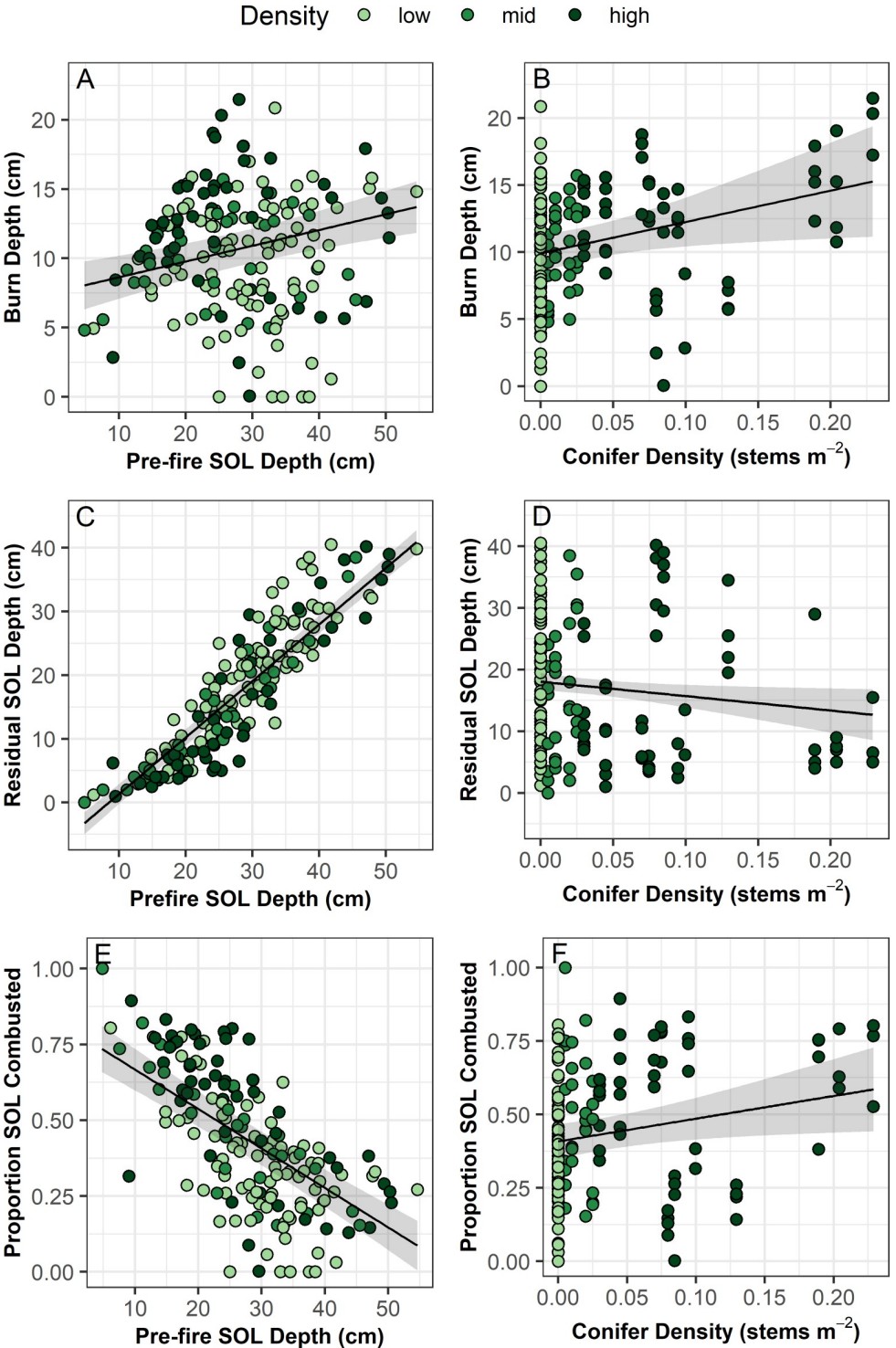

**Fig 3. Burn depth, residual soil organic layer (SOL) depth, and proportion of pre-fire SOL combusted as a function of pre-fire SOL depth and conifer density.** Black lines represent modelled relationships from linear mixed effects models with random effects of grid (2 levels) and plot (46 levels) nested within grid. Shading represents 95% confidence regions. See Table 2 for model results.

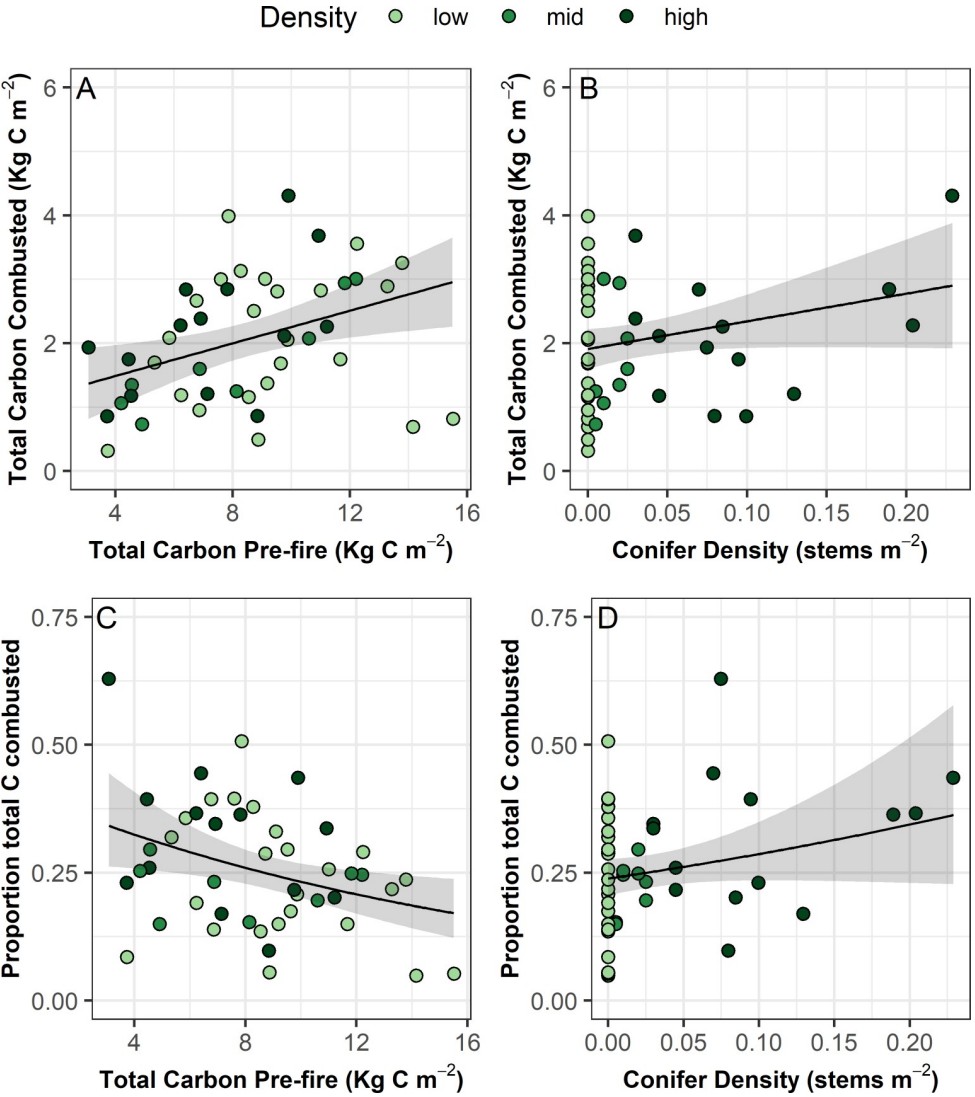

**Fig 4. Total and proportion of total C combusted as a function of total pre-fire C pool and conifer density.** Black lines represent modelled relationships and shading represents 95% confidence regions. A linear mixed effects model was used to model total C combusted and a generalized linear mixed effect model with a Gamma error distribution and log link function was used to model proportion of total C combusted. Both included a random effect of grid (2 levels). See Table 2 for model results.

## Seed rain

We estimate that seed rain in burned plots was 2.32 ± 3.06 seeds m$^{-2}$ year$^{-1}$ (Table 1). Seed rain in unburned control plots was slightly lower at 1.77 ± 2.34 seeds m$^{-2}$ year$^{-1}$. The impact of conifer density on seed rain did not differ between control and burned plots; seed rain increased in response to increasing conifer density (Fig 9 and Table 2).

## Seeding experiment

Across all treatments, we counted 883 seedlings in August 2017 and 660 in August 2018. In quadrats where at least one seedling occurred, the average number of seeds required for one seedling to germinate (i.e., the ratio of seed to seedlings) was ~83 in 2017 and ~84 in 2018. We

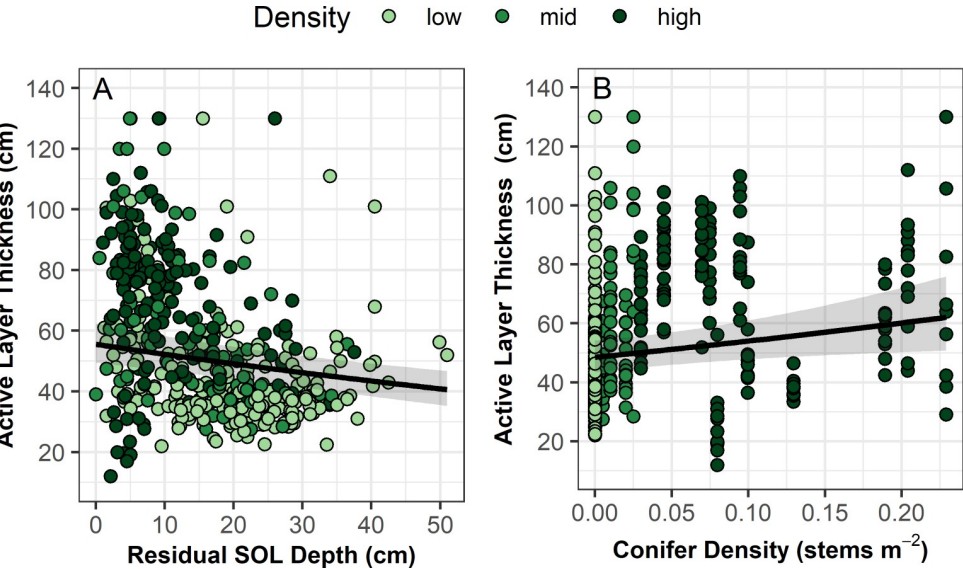

**Fig 5.** Active layer thickness as a function of a) residual SOL depth and b) pre-fire conifer density. Black line represents modelled relationships using a generalized linear mixed effect model with a Gamma error distribution and log link function and random effects of grid (2 levels) and plot (44 levels) nested within grid. Shading represents 95% confidence regions. See Table 2 for model results.

found that germination significantly increased with soil scarification and was influenced by the interaction between seed addition and whether a plot was burned or unburned (Fig 10 and Table 2). The highest seedling counts occurred in scarified and seeded quadrats (total seedling count was 567 in 2017 and 418 in 2018), closely followed by unscarred and seeded quadrats (total count was 305 in 2017 and 235 in 2018). When no seed was applied to the quadrats, seedling counts were very low (<6 seedlings in each). As we seeded approximately 250 seeds (with

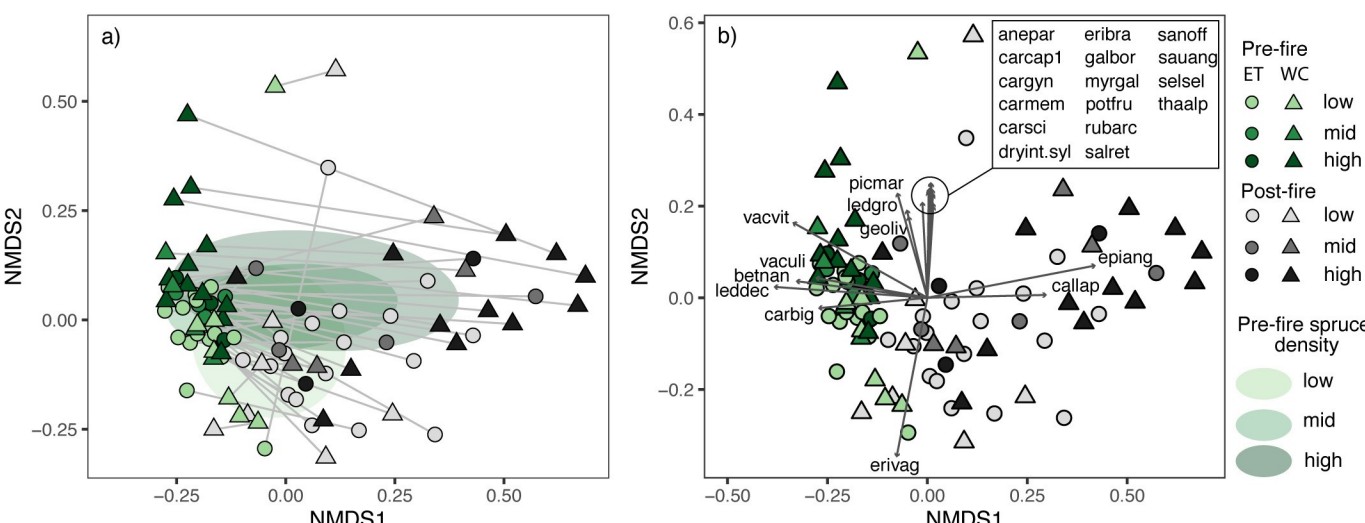

**Fig 6. Two-dimensional NMDS representation of pre- and post-fire plant community composition (Bray–Curtis distance) at each site (stress = 0.15, 48 sites, 80 species).** Points indicate a site at a given sampling time. Point colors indicate pre- (grey) and post-fire (green) plant communities and color shades indicate pre-fire spruce density. On panel a), the standard deviation ellipse is shown for each spruce density group, with the color shadings corresponding to pre-fire spruce density. Light grey lines link the same site before and after fire. On panel b), the dark grey arrows are overlays of species vectors significantly associated with plot position in the ordination space ($p \leq 0.001$). Arrow lengths were halved to facilitate visualization. Species codes can be found in S2 Table.

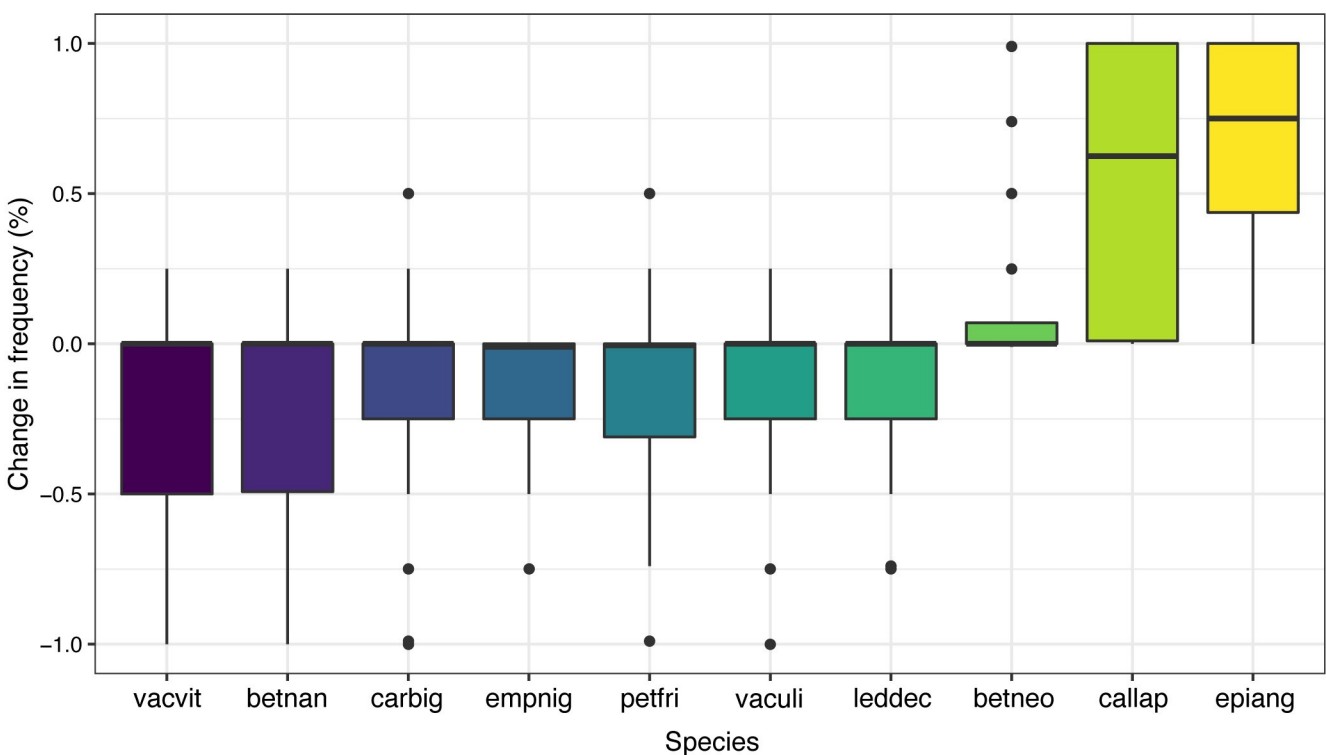

**Fig 7. Box and whiskers plot of the change in frequency of occurrence at sites pre to post-fire for the ten vascular plant species contributing the largest absolute average difference.** Species codes are the first three letters of genus and species for *Vaccinium vitis-idaea*, *Betula nana*, *Carex bigelowii*, *Empetrum nigrum*, *Petasites frigidus*, *V. uliginosum*, *Rhododendron tomentosum* subsp. *decumbens*, *B. neoalaskana*, *Calamagrostis laponica*, and *Chamerion angustifolium*.

~16% viability) per quadrat (n = 120 quadrats) the mean germination rate was 2.9% in 2017. Approximately 25% of these seedlings died, decreasing this rate to 2.2% by 2018.

## Discussion

The rate and magnitude of climate-induced shifts at the boreal forest-tundra ecotone depends on a multitude of biotic and abiotic factors including wildfire [32]. Our study spanned a gradient from tundra to low-density conifer stands, with higher density conifer sites corresponding to increased drainage and lower equivalent latitudes. Along this gradient, we saw an increase in fire severity, increased ALT, and changes in pre- and post-fire understory vegetation communities. Natural post-fire conifer tree recruitment also increased with pre-fire conifer density, suggesting that recruitment could be limited by seed availability and that areas with higher pre-fire conifer density are also areas with more suitable conditions for seedling recruitment and establishment. Our seed rain observations and seeding experiment emphasize the high recruitment potential of black spruce trees post-fire when seed is not limited and suggest that soil disturbances either from fire or other causes are important for establishment. Taken together, our results suggest that the expected climate-induced forest infilling (i.e. increased density) at the forest-tundra ecotone could increase fire severity and C emissions in a region where wildfires have historically been rare. Our observed rates of viable seed and natural seedling recruitment suggest that this infilling process was not induced by the most recent fire. However, as climate continues to warm, both the production of viable seed and disturbance from wildfires is likely to increase, which could, in turn, stimulate increases in tree recruitment and density in this region of the forest-tundra ecotone.

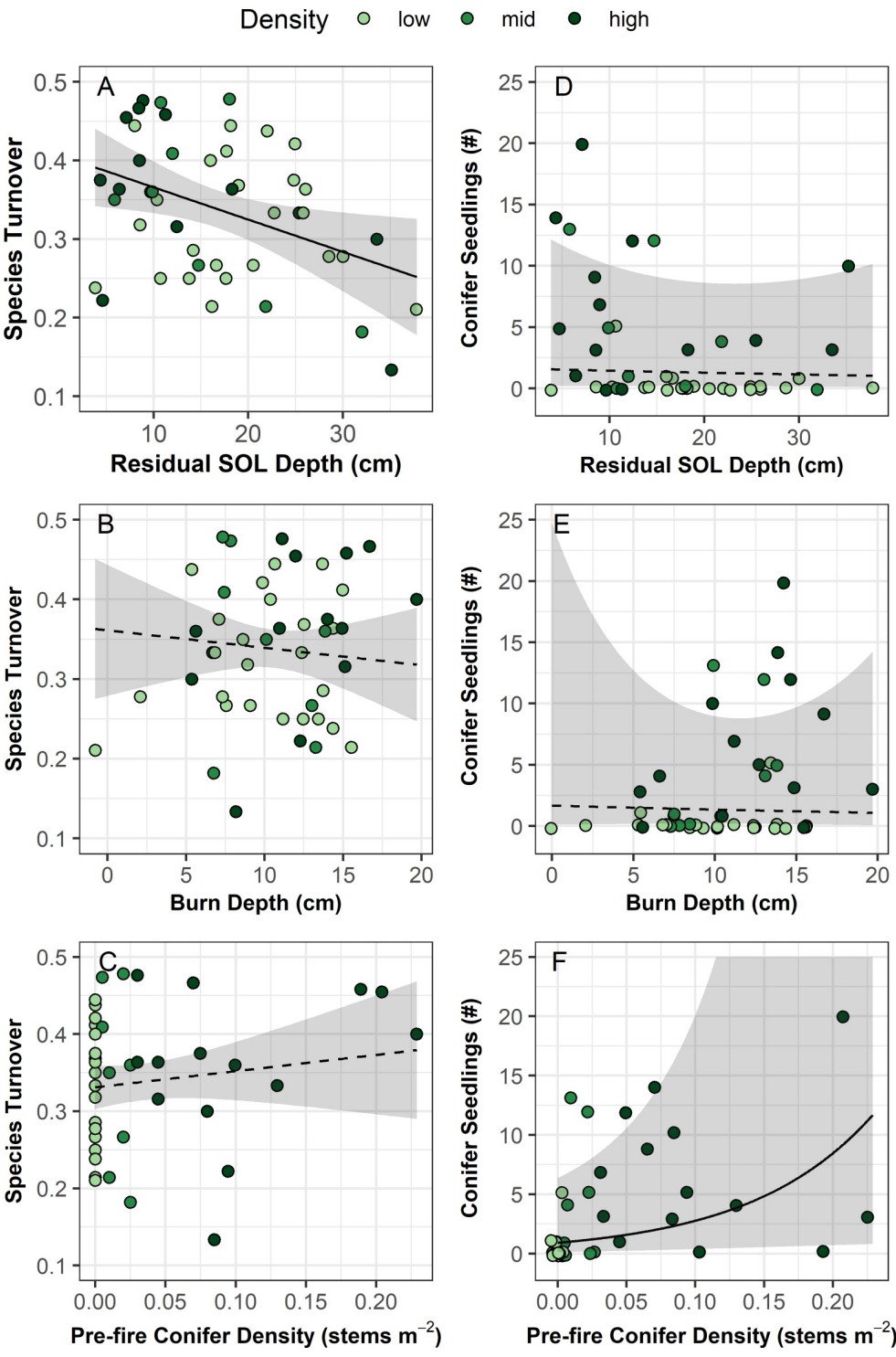

**Fig 8. Species turnover index and conifer seedling counts as a function of residual soil organic layer (SOL) depth, burn depth, and pre-fire conifer density.** Black lines represent modelled relationships (solid = significant at p-value 0.05, dashed = not significant) and shading represents 95% confidence regions. Species turnover was modelled with a linear mixed effect model and conifer seedling count was modelled with a generalized linear mixed effect model fit with a type II negative binomial distribution. Both models included a random effect of grid (2 levels). See Table 2 for model results.

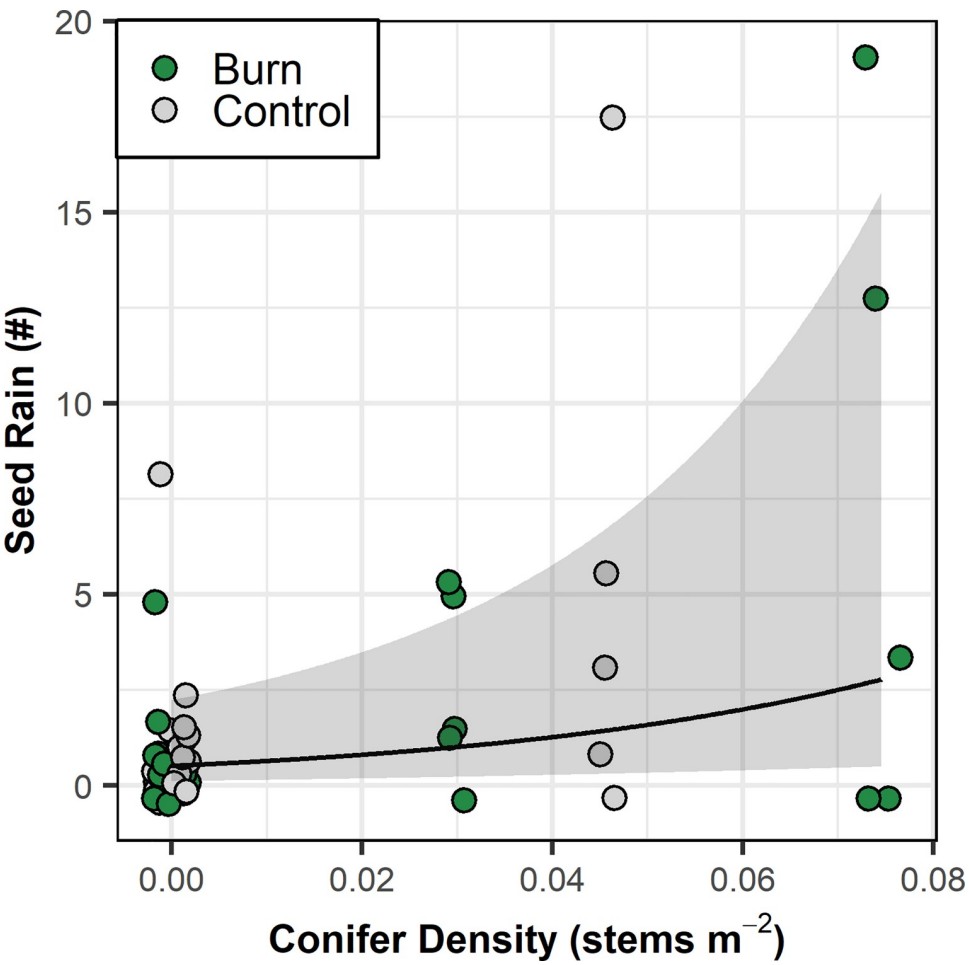

**Fig 9. Seed rain (#) as a function of conifer density (stem·m⁻²).** Black line represents modelled relationship from a generalized linear mixed effect model with a type II negative binomial distribution and random effects of collection period (5 levels) crossed with grid (2 levels). Shading represents 95% confidence region. See Table 2 for model results.

## Burn depth, carbon combustion, and active layer thickness

Similar to previous studies in northern tundra and boreal ecosystems [16,17,49], we found large variation in pre-fire SOL depth (4.8–54.6 cm) and belowground C pools (1.53–24.92 kg C m⁻²). Pre-fire SOL depth was marginally related to burn depth, but highly predictive of residual SOL, highlighting a relatively consistent burn depth (mean ± std.error = 10.57 ± 4.48) across a large range in pre-fire SOL depths (27.95 ± 9.71). This is similar to previous studies in black spruce forest of Interior Alaska [69] and the Northwest Territories, Canada [17], and tundra on the North Slope of Alaska [16] where pre-fire SOL could explain 50–90% of the variation in residual SOL depth. This finding is further supported by the strong negative relationship between proportion of SOL combusted and pre-fire SOL depth, with near complete combustion occurring when pre-fire SOL depths were shallow and relatively little proportional combustion occurring with deep pre-fire SOLs. Our data indicate that areas of low drainage with deep pre-fire SOLs are likely to burn less severely and will therefore retain a larger post-fire belowground C pool than areas of higher drainage with shallower pre-fire SOLs. These results suggest that the ecosystem function of long-term C storage is somewhat resilient to

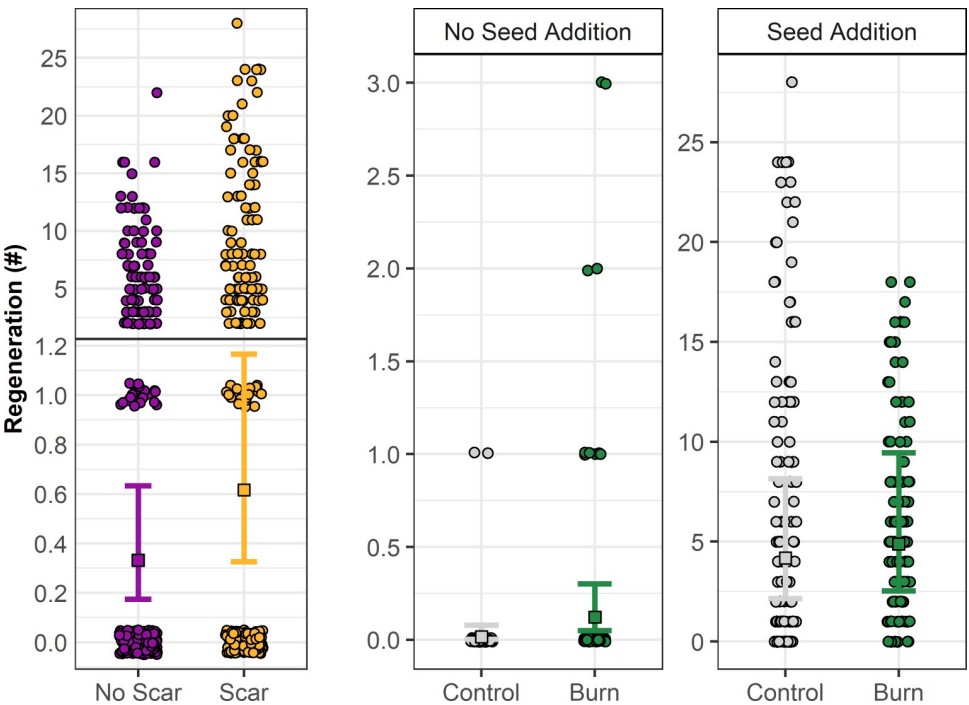

**Fig 10. Experimental spruce regeneration as a function of scarification and the interaction between burn status and seed addition.** Squares and error bars represent model fit and 95% confidence intervals from a generalized linear mixed effect model with a type I negative binomial distribution and random effects of sampling year (2 levels) crossed with grid (2 levels), plot (12 levels) nested within grid, and block (5 levels) nested within plot and grid. Points represent raw data. Note the break in Y-axis for scarification treatment and difference in Y-axes between panels. See Table 2 for model results.

disturbance from wildfires in tundra dominated sites, but that treed sites are more vulnerable to functional change following wildfire in the boreal-forest tundra ecotone of this region.

On average 1.96 kg C m$^{-2}$ was combusted from belowground C pools, which is similar to the average belowground C combusted from a tundra fire on the North Slope of Alaska (1.5 kg C m$^{-2}$) [16]. Total C combustion across all plots was on average 2.18 kg C m$^{-2}$, which is similar to the 2.02 kg C m$^{-2}$ estimated from a tundra fire on the North Slope of Alaska [16] and within the range reported from black spruce sites in Alaska (Table 3 in [24]). In this study, we did not estimate C combusted from plant biomass, but only estimated combustion from trees, soils, and moss. Mack *et al*. [16] estimated that approximately 0.5 kg C m$^{-2}$ was combusted from tundra plants, primarily tussocks of *Eriophorum vaginatum*, on the North Slope of Alaska. Near Healy, Alaska just outside of DNP, total aboveground vascular plant biomass ranges from ~0.25 to 0.38 kg C m$^{-2}$ [70]. Thus, the total aboveground C and C combustion estimates we present in this study are likely underestimated by ~ 0.2 to 0.5 kg C m$^{-2}$.

Our study spanned a gradient from tundra to low-density spruce stands along which we saw an increase in fire severity. Specifically, burn depth, total C combusted, and the proportion of total C combusted increased and residual SOL depth decreased with increasing tree density. Higher severity fires in forested compared to treeless plots might be due to increased fuel for combustion and trees increasing the rate of spread and intensity of fires. Higher tree density is also associated with increased moss cover in relatively drier landscape positions compared to tussock dominated or treeless sites. A feather moss ground layer is highly flammable under dry conditions [71] and could explain the increased burn depth associated with higher tree density.

Wildfires increase soil temperatures and result in a deeper active layer, but the magnitude of these changes is modulated by site-specific characteristics in vegetation, residual SOL depth, albedo, and drainage conditions [72]. Presumably, ALT was greater pre-fire at treed versus tundra sites based on their warmer landscape position and higher drainage conditions. Unfortunately, we do not have pre-fire ALT measurements to assess the degree of change in due to fire. Post-fire, we found that ALT increased with pre-fire conifer density and with reduced residual SOL depth. A reduction in SOL thickness increases soil thermal conductivity and temperature, and is often considered the most important factor controlling ALT and permafrost recovery after fires [73–75]. However, we found that pre-fire conifer density had a larger impact on ALT than residual SOL depth. It is well established that ecosystem properties impact ALT, with permafrost stability and rates of recovery being generally greater in poorly drained tundra environments compared to drier and warmer boreal forests ecosystems [74,76].

## Understory vegetation composition

In boreal forest ecosystems, initial post-fire plant composition is determined by a combination of disturbance and environment drivers [5]. Here, we observed large post-fire changes in plant community composition at all sites, with greater species turnover at treed sites compared to tundra sites. However, about half of the species showed little change in their post-fire frequency, which suggested that, while species' cover may be altered by fire as shown in the ordination, species occurrence at these sites was relatively resilient. Species turnover increased in sites with low residual SOL, highlighting the importance of substrate availability for the recruitment and establishment of new species following a disturbance. Environmental drivers, fire characteristics, and legacies from past vegetation were all important for determining post-fire community assembly in this region. Treed sites were generally drier than the treeless tundra sites, supporting that drainage is a key determinant of plant community composition and post-fire community assembly in boreal forests [27,28,77]. Specifically, poorly drained sites were dominated by rapidly re-sprouting tussock sedges (*E. vaginatum*) or ericaceous shrubs and thus exhibited less change in species composition compared to treed sites that were dominated by an extensive feather moss ground layer pre-fire. Combustion of this moss layer, paired with the removal of the tree canopy likely enabled the colonization of small-seeded, wind dispersed, and shade intolerant species, such as the deciduous tree *B. neoalaskana*, fireweed (*C. angustifolium*), or the grass *C. lapponica*. In contrast, light levels would not have changed in the treeless sites, allowing the reestablishment of plant community more similar to pre-fire composition.

Although all sites had observable changes in species composition, there were no obvious shifts of tundra sites becoming more similar to treed sites or the reverse. Compositional turnover was overall relatively low at our sites, indicating that landscape vegetation mosaics ordered by topography-driven environmental gradients may persist across fire disturbances. Our sampling took place only three years after fire and if pre-fire spruce canopies are not reestablished then understory communities might resemble those of the treeless tundra sites in the future, with short-statured ericaceous shrubs and tussock sedge dominated plant communities.

## Spruce regeneration

Successful regeneration of boreal tree species requires suitable climatic conditions: both pre-fire for viable seed production and post-fire for suitable microsites for germination and establishment [36] and survival and growth [78,79]. Fires that reduce the residual SOL and expose mineral soil create microsites suitable for spruce regeneration [40,80]. In contrast, we found

that post-fire spruce regeneration was not influenced by burn depth or residual SOL depth. This might be partially due to residual SOLs still being relatively deep post-fire (mean ± sd = 16.06 ± 9.83, n = 470) or other environmental factors affecting regeneration that we did not assess. Our seeding experiment suggests that both fire and soil disturbance are important for regeneration, with higher establishment occurring in burned relative to control plots and with soil scarification relative to no scarification.

The number of naturally occurring spruce seedlings increased with pre-fire conifer density. This suggests that both topo-edaphic landscape position (pre-fire conifer density is higher in drier and warmer landscapes) and seed availability are important constraints on boreal tree recruitment in this region of the forest-tundra ecotone. Our seed trap collections and germination trials (S1 File) confirm that there is a lack of viable seeds for substantial spruce regeneration. Specifically, seed rain was approximately 2.3 seeds $m^{-2}$ $year^{-1}$ across all sites. This was highest in the higher pre-fire spruce density sites, but when there were no trees pre-fire, seed rain was approximately zero. Our laboratory germination trials resulted in an approximate viability of 16% (S1 File), meaning that on average viable seed rain was ~0.3 seeds $m^{-2}$ $year^{-1}$, an amount that could potentially replace pre-fire spruce densities, but is insufficient for infilling or expansion. Note that our seed trap collection in burned plots occurred three years post-fire and thus we likely missed the pulse of black spruce seed rain from semi-serotinous cones that occur in the first two years following fire [81]. In DNP, white spruce produced a moderate to good cone crop in the masting year 2016, but cone production was very low in 2017 and 2018 (Roland, *pers. comm.*). We did not differentiate between black and white spruce seeds in the seed traps, but there were white spruce present at these sites and in the general vicinity. Because white spruce does not require fire to release seed, seeds may have dispersed from outside the fire scar, suggesting that distance to unburned edge might also influence seed rain. However, sites with the greatest seed rain were actually located in the interior of the fire scar, not at the edge suggesting that seed dispersion from outside the fire scar was minimal.

Our seed rain values are lower than previous studies in black spruce forest in Interior Alaska [38,82,83] and for white spruce treeline sites in DNP [42]. Johnstone *et al*. [83] sampled one year after fire and found 4.76–276 seeds $m^{-2}$ $year^{-1}$ and 0–60 viable seeds $m^{-2}$ $year^{-1}$. Zasada *et al*. [82] sampled 2–3 years post fire, but still report much higher seed (41–77 seeds $m^{-2}$ $year^{-1}$) and viable seed (28–47 seeds $m^{-2}$ $year^{-1}$) than we found. Brown and Johnstone [38] conducted a seeding experiment sowed at the same seeding rates as our study, but with seeds half as viable (~7.2%) and found much greater first year regeneration across all treatments than observed in our study. Each of these studies determined that a primary driver of seed rain was black spruce basal area. An average (mean ± se) of 338 ± 43 and 18 ± 2 white spruce seeds $m^{-2}$ in forested and treeline sites respectively, have been observed in DNP [42]. Climatic conditions in the years preceding dispersal are important for the quantity and viability of both black spruce [36] and white spruce [42] seeds. In DNP, low growing degree days at high-elevation sites can strongly diminish white spruce seed availability and viability in treeline stands [42]. Thus, the relatively low rates of seed rain that we observed are most likely a product of both very low pre-fire conifer density, with many sites having no trees per-fire, and environmental conditions that limit viable seed production.

Our experimental addition of seeds to post-fire environments further highlights the limiting effect of seed availability on boreal tree recruitment. Regardless of scarification in both control and burn plots, the greatest probability of germination occurred with seed addition. The effect of seeding was almost nine times the effect of scarification and three times the effect of burning on black spruce regeneration. These results corroborate findings from Brown *et al*. [80], who show that seed limitation is the primary control of successful post-fire regeneration of boreal tree species in Interior Alaska.

## Conclusions

Understanding the impacts of wildfire throughout the forest-tundra ecotone is required to make predictions of the rate and magnitude of changes in boreal-tundra landcover, its future flammability, and associated feedbacks to the global C cycle and climate. Sites located in warmer and drier landscape positions that support a higher density of trees experienced greater fire severity and C emissions compared to sites in cool and wet landscapes with very few conifer trees. These findings suggest that treed areas within the forest-tundra ecotone are more vulnerable to wildfire induced losses of long-term C storage than treeless tundra areas. Forested sites also had higher ALT post-fire and experienced the largest fire-induced changes in understory vegetation communities. However, we did not observe strong shifts from tundra to treed sites, or the reverse. We conclude that if forest infilling (i.e., increased density) occurs throughout the forest-tundra ecotone, increases in wildfire severity and C emissions, deepening of the ALT, and changes in understory vegetation communities are also likely to occur. However, our observation of low seed availability and natural recruitment combined with our experimental results indicate that infilling has not occurred with the most recent fire and is unlikely to occur in the future without increases in the availability of viable seed.

## Supporting information

**S1 Table. Model results for soil carbon content.** Results of linear mixed-effects model of the cumulative sum of carbon content (kg·C·m$^{-2}$) as a function of depth, depth$^2$, moisture class, and the interaction between depth and moisture class. The model was fit on 205 soil increments from 50 soil monoliths in 15 plots. Random effects of plot and soil monolith nested within plot were used to account for the depth non-independence of soil depth increments sampled within a soil monolith and the spatial non-independence of soil monoliths located within a plot.
(DOCX)

**S2 Table. Understory plant composition.** Number of sites where we observed each species and their average frequency (%) pre-fire, post-fire, and in total. Nomenclature follows Hulten (1968)* and other sources, with current accepted Integrated Taxonomic System nomenclature indicated in square brackets (ITIS 2021)**.
(DOCX)

**S1 File. Seed collection and germination trials.** Detailed methods and results for seed collections and germination trials.
(DOCX)

## Acknowledgments

We thank Sarah Stein for her help and expertise conducting fieldwork in Denali National Park and Preserve. We thank our lab members at Northern Arizona University for their input and feedback at various stages of this manuscript. We extend our appreciation to the numerous field and laboratory assistants and graduate students from Northern Arizona University. The authors thank Denali National Park and Preserve and the NPS Inventory and Monitoring Program for logistical support of this project.

## Author Contributions

**Conceptualization:** Xanthe J. Walker, Carl Roland, Edward A. G. Schuur, Michelle C. Mack.

**Data curation:** Xanthe J. Walker, Brain K. Howard, Mélanie Jean, Carl Roland, Brendan M. Rogers, Kylen K. Solvik.

**Formal analysis:** Xanthe J. Walker, Mélanie Jean.

**Funding acquisition:** Jill F. Johnstone, Edward A. G. Schuur, Michelle C. Mack.

**Investigation:** Xanthe J. Walker, Brain K. Howard, Mélanie Jean, Carl Roland, Brendan M. Rogers, Michelle C. Mack.

**Methodology:** Xanthe J. Walker, Jill F. Johnstone, Carl Roland, Brendan M. Rogers, Kylen K. Solvik, Michelle C. Mack.

**Project administration:** Xanthe J. Walker, Michelle C. Mack.

**Supervision:** Xanthe J. Walker, Michelle C. Mack.

**Validation:** Xanthe J. Walker.

**Visualization:** Xanthe J. Walker, Mélanie Jean, Brendan M. Rogers.

**Writing – original draft:** Xanthe J. Walker, Mélanie Jean.

**Writing – review & editing:** Xanthe J. Walker, Brain K. Howard, Mélanie Jean, Jill F. Johnstone, Carl Roland, Brendan M. Rogers, Edward A. G. Schuur, Michelle C. Mack.

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
