## [Decision Letter · Decision Letter 0]

1 Sep 2021

PONE-D-21-20614

Impacts of pre-fire conifer density and wildfire severity on ecosystem structure and function at the forest-tundra ecotone

PLOS ONE

Dear Dr. Walker,

Thank you for submitting your manuscript to PLOS ONE. After careful consideration, we feel that it has merit but does not fully meet PLOS ONE’s publication criteria as it currently stands. Therefore, we invite you to submit a revised version of the manuscript that addresses the points raised during the review process.

We look forward to receiving your revised manuscript.

Kind regards,

Dafeng Hui, Ph.D.

Academic Editor

PLOS ONE

Journal Requirements:

This project was supported by funding awarded to MCM, JFJ and EAGS from the NASA Arctic Boreal and Vulnerability Experiment (ABoVE) Legacy Carbon grant NNX15AT71A and the Bonanza Creek Long-Term Ecological Research Program, which is funded by National Science Foundation grant # DEB1636476 and the USDA Forest Service Pacific Northwest Research Station. 

This project was supported by funding awarded to MCM, JFJ and EAGS from the NASA Arctic Boreal and Vulnerability Experiment (ABoVE) Legacy Carbon grant NNX15AT71A and the Bonanza Creek Long-Term Ecological Research Program, which is funded by National Science Foundation grant # DEB1636476 and the USDA Forest Service Pacific Northwest Research Station.

This project was supported by funding awarded to MCM, JFJ and EAGS from the NASA Arctic Boreal and Vulnerability Experiment (ABoVE) Legacy Carbon grant NNX15AT71A and the Bonanza Creek Long-Term Ecological Research Program, which is funded by National Science Foundation grant # DEB1636476 and the USDA Forest Service Pacific Northwest Research Station.

Additional Editor Comments:

I now have one report from an expert reviewer who is general positive on the manuscript, but also raised some technique concerns that need to be addressed.

Reviewers' comments:

Reviewer's Responses to Questions

**Comments to the Author**

1. Is the manuscript technically sound, and do the data support the conclusions?

Reviewer #1: Yes

2. Has the statistical analysis been performed appropriately and rigorously? 

Reviewer #1: Yes

3. Have the authors made all data underlying the findings in their manuscript fully available?

Reviewer #1: Yes

4. Is the manuscript presented in an intelligible fashion and written in standard English?

Reviewer #1: Yes

5. Review Comments to the Author

Reviewer #1: This study examined the potential impacts of wildfire severity on understory vegetation composition, conifer recruitment and ALT across tundra-boreal forest ecotone based on field observations and experiments. The manuscript is well written and professionally presented. The sampling strategies and statistical analyses are carefully designed and sound. I recommend the publication of this manuscript with minor modifications and clarification. Yet both the page and row numbers are missing in the submitted manuscript, which makes the review process a bit inconvenient. So here I provide my comments based on page numbers of the PDF file generated by the journal system.

1) Page 10 of the generated pdf file: “Treeless tundra ecosystems are generally wetter with thicker SOL layers than forested ecosystems.” Not necessarily thicker SOL in tundra. Measurements from several previous studies reported thicker SOL in boreal forests than in tundra (a few examples below). Please justify and provide references for your statement.

Baughman, C.A., Mann, D.H., Verbyla, D.L., Kunz, M.L., 2015. Soil surface organic layers in Arctic Alaska: Spatial distribution, rates of formation, and microclimatic effects. J. Geophys. Res. G Biogeosciences 120, 1150–1164.

Harris, S.A., 1987. Influence of organic (Of) layer thickness on active-layer thickness at two sites in the western Canadian Arctic and Subarctic. Erdkunde 41, 275–285.

Jiang, Y., Rocha, A. V, O’Donnell, J. a, Drysdale, J. a, Rastetter, E.B., Shaver, G.R., Zhuang, Q., 2015. Contrasting soil thermal responses to fire in Alaska tundra and boreal forest. J. Geophys. Res. Earth Surf. 1–16.

Kasischke, E.S., Johnstone, J.F., 2005. Variation in postfire organic layer thickness in a black spruce forest complex in interior Alaska and its effects on soil temperature and moisture. Can. J. For. Res. 35, 2164–2177.

Lafleur, B., Cazal, A., Leduc, A., Bergeron, Y., 2015. Soil organic layer thickness influences the establishment and growth of trembling aspen (Populus tremuloides) in boreal forests. For. Ecol. Manage. 347, 209–216.

2) First paragraph on page 13 of the pdf file: The format of the references should be consistent throughout the manuscript. Here “(Roland et al., 2013)” should be changed to [41]. Some other references in this manuscript need to be modified as well, such as “(Roland et al., 2014)”.

3) Second paragraph on page 13: Though the text mentioned that the control plots were randomly selected, they do not seem to be randomly distributed in space according to Figure 1. How do you weight the cover types by their distribution?

4) Third paragraph on page 13: How do you determine the moisture categories for the plots? Here the text mentioned that there are four categories, while Figure 2 only shows three. Also, the two subplots were not properly labeled as a) or b) in Figure 2.

5) “Burn depth, carbon combustion, and active layer thickness” in “Data Analysis” (page 18) and “Discussion” (page 25) sections:

The authors modeled combusted soil C using LMMs to predict combusted C and calculate total belowground C for all sites. The manuscript only provides the coefficients and p-values for independent variables for the LMMs, while the overall performances of the model are missing. Can the LMMs predict the combusted soil C well? Since the belowground C contributes to the total C much more the aboveground C in these sites, the total C is largely affected by whether the LMMs can predict the combusted soil C accurately. And the findings regarding pre-fire conifer density impacts on total C will be questionable.

6) Data availability on page 8: The github link seems not working. Please double check.

6. PLOS authors have the option to publish the peer review history of their article (what does this mean?). If published, this will include your full peer review and any attached files.

Reviewer #1: No

---

## [Author Response · Author response to Decision Letter 0]

28 Sep 2021

Response to Reviews

Please find enclosed each of the comments made by the editor and the reviewer and our responses. 

We have ensured that the manuscript meets the style requirements. 

This project was supported by funding awarded to MCM, JFJ and EAGS from the NASA Arctic Boreal and Vulnerability Experiment (ABoVE) Legacy Carbon grant NNX15AT71A and the Bonanza Creek Long-Term Ecological Research Program, which is funded by National Science Foundation grant # DEB1636476 and the USDA Forest Service Pacific Northwest Research Station. Please state what role the funders took in the study. If the funders had no role, please state: "The funders had no role in study design, data collection and analysis, decision to publish, or preparation of the manuscript." 

We have included the following statement in our cover letter: “The funders had no role in study design, data collection and analysis, decision to publish, or preparation of the manuscript.”

This project was supported by funding awarded to MCM, JFJ and EAGS from the NASA Arctic Boreal and Vulnerability Experiment (ABoVE) Legacy Carbon grant NNX15AT71A and the Bonanza Creek Long-Term Ecological Research Program, which is funded by National Science Foundation grant # DEB1636476 and the USDA Forest Service Pacific Northwest Research Station. Please note that funding information should not appear in the Acknowledgments section or other areas of your manuscript. We will only publish funding information present in the Funding Statement section of the online submission form. 

Please remove any funding-related text from the manuscript and let us know how you would like to update your Funding Statement. Currently, your Funding Statement reads as follows: This project was supported by funding awarded to MCM, JFJ and EAGS from the NASA Arctic Boreal and Vulnerability Experiment (ABoVE) Legacy Carbon grant NNX15AT71A and the Bonanza Creek Long-Term Ecological Research Program, which is funded by National Science Foundation grant # DEB1636476 and the USDA Forest Service Pacific Northwest Research Station.

We have removed the funding from our acknowledgements. Lines 687-698 now read:

“We thank Sarah Stein for her help and expertise conducting fieldwork in Denali National Park and Preserve. We thank our lab members at Northern Arizona University for their input and feedback at various stages of this manuscript. We extend our appreciation to the numerous field and laboratory assistants and graduate students from Northern Arizona University. The authors thank Denali National Park and Preserve and the NPS Inventory and Monitoring Program for logistical support of this project.”

The amended funding statement should read:

“This project was supported by funding awarded to MCM, JFJ and EAGS from the NASA Arctic Boreal and Vulnerability Experiment (ABoVE) Legacy Carbon grant NNX15AT71A and the Bonanza Creek Long-Term Ecological Research Program, which is funded by National Science Foundation grant # DEB1636476 and the USDA Forest Service Pacific Northwest Research Station. The funders had no role in study design, data collection and analysis, decision to publish, or preparation of the manuscript.”

The data and code is now available online and this is included on the first page of the manuscript submission. The data availability statement on lines 30-38 now reads:

“All data and R code from this study is archived in the US National Science Foundation-funded Bonanza Creek Long Term Ecological Research Data Catalog, which is part of EDI Data Portal. R code used for statistical analyses is also archived on github at https://github.com/xanthewalker/Denali_fire

Walker, Xanthe; Mack, Michelle C; Johnstone, Jill. 2021. Toklat River Fire in Denali National Park and Preserve: Site level environmental, soil, tree, vegetation, and fire characteristics measured in 2016, Bonanza Creek LTER - University of Alaska Fairbanks. BNZ:786, http://www.lter.uaf.edu/data/data-detail/id/786”

We do not believe there are any copyright issues with this figure since all the data layers shown are publicly-available online and not proprietary: Alaska Large Fire Database, Denali National Park outline, North American roads, and tree cover from Sexton et al 2013. 

Supporting information is now listed at the end of the main document. 

We have reviewed the reference list and we believe that it is complete and correct. 

Additional Editor Comments:

I now have one report from an expert reviewer who is general positive on the manuscript, but also raised some technique concerns that need to be addressed.

Reviewers' comments:

Reviewer's Responses to Questions

Comments to the Author

5. Review Comments to the Author

Reviewer #1: This study examined the potential impacts of wildfire severity on understory vegetation composition, conifer recruitment and ALT across tundra-boreal forest ecotone based on field observations and experiments. The manuscript is well written and professionally presented. The sampling strategies and statistical analyses are carefully designed and sound. I recommend the publication of this manuscript with minor modifications and clarification. Yet both the page and row numbers are missing in the submitted manuscript, which makes the review process a bit inconvenient. So here I provide my comments based on page numbers of the PDF file generated by the journal system.

1) Page 10 of the generated pdf file: “Treeless tundra ecosystems are generally wetter with thicker SOL layers than forested ecosystems.” Not necessarily thicker SOL in tundra. Measurements from several previous studies reported thicker SOL in boreal forests than in tundra (a few examples below). Please justify and provide references for your statement.

We have removed that sentence and amended the text as follows. Lines 85 to 91 now read: “Field based estimates of tundra fire severity are rare, but pre-fire C pools and C emissions per unit area reported for a tundra wildfire in Alaska [16] are at the high and low range, respectively, of those reported for boreal black spruce forests [23,24]. As such, the severity of wildfires and C emissions might decrease in association with decreasing tree cover and increased SOL across the forest-tundra ecotone.”

2) First paragraph on page 13 of the pdf file: The format of the references should be consistent throughout the manuscript. Here “(Roland et al., 2013)” should be changed to [41]. Some other references in this manuscript need to be modified as well, such as “(Roland et al., 2014)”.

Thank you for pointing out this error. We have formatted the references. 

3) Second paragraph on page 13: Though the text mentioned that the control plots were randomly selected, they do not seem to be randomly distributed in space according to Figure 1. How do you weight the cover types by their distribution?

We have changed the text to clarify how our sampling and choosing of the control plots was completed. Lines 178 to 185 now read:

“An additional nine control plots were established ~40 km east, along Stampede Road near Healey, AK, to calibrate our methods for assessing burn depth. Control plots were selected through a GIS approach using the DNP Land Cover Map. Cover types within the burned study sites were weighted by their distribution and then within these selected cover types, tree cover and aspect were selected from their weighted distributions. Control plots were then chosen to have the same landcover within ± 3% tree cover, with matching slopes and aspects to conditions characterized within the burned monitoring plots.”

4) Third paragraph on page 13: How do you determine the moisture categories for the plots? Here the text mentioned that there are four categories, while Figure 2 only shows three. Also, the two subplots were not properly labeled as a) or b) in Figure 2.

Thank you for pointing out that this was not clear. We have changed the text to clarify that methods for describing moisture category are outline in the Johnstone et al. 2008 and that although we originally had 4 moisture categories, this was changed to three for subsequent analysis due to the low representation in the driest category. 

The text on lines 189-192 now read: 

“Soil monoliths and topographic factors were used to assign a potential site moisture class, as described in [46]. Plots ranged from subxeric-mesic to subhygric among four categories. Only two burned plots were in the subxeric-mesic category, and we therefore reclassified them as mesic for subsequent analyses, resulting in three moisture categories.”

5) “Burn depth, carbon combustion, and active layer thickness” in “Data Analysis” (page 18) and “Discussion” (page 25) sections:

The authors modeled combusted soil C using LMMs to predict combusted C and calculate total belowground C for all sites. The manuscript only provides the coefficients and p-values for independent variables for the LMMs, while the overall performances of the model are missing. Can the LMMs predict the combusted soil C well? Since the belowground C contributes to the total C much more the aboveground C in these sites, the total C is largely affected by whether the LMMs can predict the combusted soil C accurately. And the findings regarding pre-fire conifer density impacts on total C will be questionable.

We have now included an additional column in table 2 which has marginal (only fixed effects) and conditional (fixed and random effect) R2 values showing how much variance was explained by each of the models.

6) Data availability on page 8: The github link seems not working. Please double check.

Thank you for pointing this out. We have changed the link and we have also archived our code with the rest of the data at the Bonanza Creek Long Term Ecological Research Data Catalog.

---

## [Editor Report · Decision Letter 1]

30 Sep 2021

Impacts of pre-fire conifer density and wildfire severity on ecosystem structure and function at the forest-tundra ecotone

PONE-D-21-20614R1

Dear Dr. Walker,

We’re pleased to inform you that your manuscript has been judged scientifically suitable for publication and will be formally accepted for publication once it meets all outstanding technical requirements.

Kind regards,

Dafeng Hui, Ph.D.

Academic Editor

PLOS ONE

Additional Editor Comments (optional):

The authors have adequately address the reviewer's concerns.
---

## [Editor Report · Acceptance letter]

8 Oct 2021

PONE-D-21-20614R1 

Impacts of pre-fire conifer density and wildfire severity on ecosystem structure and function at the forest-tundra ecotone 

Dear Dr. Walker:

I'm pleased to inform you that your manuscript has been deemed suitable for publication in PLOS ONE. Congratulations! Your manuscript is now with our production department. 

Kind regards, 

on behalf of

Dr. Dafeng Hui 

Academic Editor

PLOS ONE